# Strengthening Primary Health Care Through Implementation Research: Strategies for Reaching Zero-Dose Children in Low- and Middle-Income Countries’ Immunization Programs

**DOI:** 10.3390/vaccines13101040

**Published:** 2025-10-09

**Authors:** Boniface Oyugi, Karin Kallander, A. S. M. Shahabuddin

**Affiliations:** 1Maternal Newborn Child and Adolescent Health Section, Health Programme Group, United Nations Children’s Fund, New York, NY 10017, USA; boyugi@unicef.org (B.O.); kkallander@unicef.org (K.K.); 2Centre for Health Services Studies (CHSS), University of Kent, George Allen Wing, Canterbury CT2 7NF, UK; 3Global Child Health and The Sustainable Development Goals Group, Department of Global Public Health, Karolinska Institutet, Tomtebodavägen 18 A, 171 77 Stockholm, Sweden

**Keywords:** immunization inequity, zero-dose, implementation research, primary health care, LMICs

## Abstract

**Introduction:** Despite global improvements in immunization, major gaps persist. By 2024, an estimated 14.3 million infants, predominantly in low- and middle-income countries (LMICs), remained zero-dose (ZD), never having received even the first DTP vaccine. In 2022, 33 million children missed their measles vaccination (22 million missed the first dose, 11 million missed the second dose), highlighting entrenched structural, behavioral, and systemic barriers that continue to exclude marginalized populations. Addressing these inequities requires innovative, context-adapted approaches that strengthen primary health care (PHC) and extend services to the hardest-to-reach populations. **Objectives:** This study aims to document and synthesize implementation research (IR) projects on immunization programs in LMICs, identifying key enablers and effective strategies that reduce inequities, improve outcomes, and support efforts to reach ZD children. **Methods:** We conducted a retrospective multiple-case study of 36 IR projects across 13 LMICs, embedded within an evidence review framework and complemented by policy analysis. Data were drawn from systematic document reviews and validation discussions with project leads. A total of 326 strategies were extracted, coded using a structured codebook, and mapped to the WHO–UNICEF PHC Levers for Action. Descriptive analysis synthesized patterns across service delivery and policy outcomes, including coverage gains, improved microplanning, community engagement, and system integration. **Results:** Of the 326 immunization strategies identified, most (76.1%) aligned with operational PHC levers, particularly monitoring and evaluation (19.3%), workforce development (18.7%), and models of care (12%). Digital technologies (11.7%) were increasingly deployed for real-time tracking and oversight. Core strategic levers comprised 23.9% of strategies, with community engagement (8.9%) and governance frameworks (7.7%) emerging as critical enablers, though sustainable financing (4%) and private-sector engagement (0.9%) were rarely addressed. While the majority of projects focused on routine immunization (n = 32), only a few directly targeted ZD children (n = 3). Interventions yielded improvements in both service delivery and policy outcomes. Improvements in microplanning and data systems (23.5%) reflected the increased uptake of digital dashboards, GIS-enabled tools, and electronic registries. Community engagement (16.2%) emphasized the influence of local leaders and volunteers in building trust, while health system strengthening (15.7%) invested in cold chain, supervision, and workforce capacity. Coverage gains (10.6%) were achieved through delivery innovations, though sustainable financing remained a critical problem (3.4%). **Conclusions:** Reaching ZD children requires equity-driven strategies that combine digital innovations, community engagement, and resilient system planning. Sustained progress depends on strengthening governance, financing, and research. Embedding IR in immunization programs generates actionable evidence, supports context-specific strategies, and reduces equity gaps, offering practical insights that complement health system research and advance the Immunization Agenda 2030.

## 1. Introduction

As of 2024, childhood immunization remains one of the most cost-effective public health strategies globally [1]. Over the past five decades, immunization efforts have averted an estimated 154 million deaths, 146 million of which were among children under five [1]. Despite these remarkable gains, immunization coverage gaps persist, particularly in low-income and lower-middle-income countries (LICs and LMICs). These gaps are most starkly reflected among zero-dose (ZD) children, defined as those who have not received even a single dose of the diphtheria–tetanus–pertussis (DTP) vaccine [2,3,4]. In 2022, an estimated 33 million children missed at least one dose of the measles vaccine—22 million missed their first dose and 11 million missed their second dose [5]. By 2024, approximately 14.3 million infants worldwide had received no vaccines at all, classifying them as ‘zero-dose’ children [6].

Despite economic gains in many LMICs, deep-seated structural inequities continue to marginalize vulnerable populations, with ZD children disproportionately found in urban slums, remote rural areas, and conflict-affected zones. Children in the poorest wealth quintile are four times more likely to be ZD children than those in the wealthiest quintile [7,8], positioning ZD status as a critical marker of broader health and social exclusion [3]. Addressing these disparities is fundamental to achieving the Immunization Agenda 2030 (IA2030) and the Sustainable Development Goals (SDGs), which emphasize universal access to essential services and the reduction in preventable child deaths [9].

To tackle these inequities, WHO has underscored primary health care (PHC) as the foundation for universal health coverage (UHC), emphasizing equitable, community-based service delivery [10]. Aligned with this vision, IA2030 targets a 50% reduction in the number of ZD children by 2030 through strategies that embed immunization into broader PHC systems and strengthen subnational capacity for both routine and outreach services [9,11]. UNICEF plays a pivotal role in reaching ZD children by supporting national efforts, strengthening health systems, and expanding immunization access in the most underserved and high-risk communities globally [12]. Guided by the Equity Reference Group for Immunization, UNICEF focuses its efforts on four high-burden contexts, namely remote rural areas, urban slums, conflict zones, and mobile populations, using tools such as the Equity Tracker and Demand Strategy Builder to better identify and serve these communities [13].

Importantly, ZD status is more than just an immunization metric—it signals deep, multisectoral deprivation. These children are often excluded from essential services, including health care, education, nutrition, water, and sanitation, highlighting systemic marginalization [12,14]. Meeting their needs demands trust-building, context-sensitive strategies that address both logistical and social barriers. This includes integrating essential services and adapting delivery models to local needs [15]. Achieving this requires the deliberate expansion of PHC systems, with immunization embedded as a core component, supported by flexible, locally adapted implementation models.

Implementation research (IR) is a pivotal tool in transforming immunization programs to be more equitable and effective. By generating real-time, context-specific evidence, IR empowers governments and partners to design interventions tailored to community needs, optimize resource allocation, and refine delivery and monitoring systems [16,17,18,19]. This approach is particularly critical in fragile, conflict-affected, and mobile settings, where conventional service models often break down. In such contexts, IR facilitates the co-creation of adaptive, scalable, and resilient solutions rooted in community engagement and systemic learning [20].

UNICEF has placed IR at the center of its strategy to reduce immunization inequities across LMICs. Its IR portfolio spans key domains, including immunization, maternal and child health, WASH, nutrition, digital health, and education, generating actionable insights from resource-constrained settings [17,21]. Since 2014, UNICEF, through its Primary Health Care and Health Systems Strengthening Unit (PHC-HSS Unit), has played a central role in advancing IR to strengthen the Expanded Programme on Immunization (EPI) across more than 15 countries. With over 40 IR projects supported globally [21], UNICEF has worked to dismantle implementation bottlenecks, test scalable strategies, and develop policy-relevant insights. These efforts have focused on challenges ranging from vaccine delivery and demand generation to health system integration and equity. A key aspect of UNICEF’s approach is integrating IR into national immunization programs by promoting collaboration among policymakers, researchers, and implementers [22,23,24]. Additionally, UNICEF has led multisectoral strategies to reach ZD children by integrating immunization with nutrition, WASH, and education services [25]. Its IR agenda focuses on community-driven approaches that produce actionable evidence aligned with grassroots needs and national policies. These efforts aim to improve service delivery and ensure that no child is left behind in achieving immunization equity.

This study aims to document and compile the results of the UNICEF support IR projects focusing on immunization programs in LMICs. The review identifies implementation enablers and optimal immunization strategies that have worked well in reducing immunization inequity and achieving better immunization outcomes and which can be used to reach ZD children.

## 2. Materials and Methods

### 2.1. Study Design

This study employed a retrospective, multiple-case study design [26] embedded within an evidence review framework [27,28] and complemented with policy analysis elements [29,30]. This approach was well suited for examining completed UNICEF-supported IR projects conducted across diverse LMICs. The study’s retrospective nature enabled a thorough examination of completed IR projects to identify what worked, for whom, and under which conditions, while the multiple-case study structure enabled comparative insights across different geographic and programmatic contexts [26,27]. Each case was defined as an IR project that explicitly applied IR methods to address barriers to immunization uptake. These cases were drawn from a diverse set of LMICs where UNICEF-supported IR projects had been designed and implemented [21]. The identification of the case and analysis was guided by pre-defined parameters, including the temporal span of the implementation, the target population (comprising health care providers, policymakers, and service beneficiaries), and the intended outcomes, such as improvements in vaccine coverage, policy integration, and system strengthening. The evidence review framework enabled systematic data collection and integration through structured document reviews, providing a comprehensive view of the IR landscape and its alignment with strategic priorities. This was complemented by consultative validation discussions with team leads from selected IR projects, which offered real-time interpretation and ensured coherence with operational priorities. The analysis was further strengthened via the incorporation of policy analysis elements, which linked project-level strategies to broader governance, financing, and system-level implications. Taken together, this integration of evidence review, comparative case analysis, and policy perspectives enhanced both the scientific rigor and the practical relevance of this study’s recommendations [31].

### 2.2. Analytical Framework

To guide the analysis, we used the WHO-UNICEF PHC Levers for Action [32] as the main framework for evaluating the immunization strategies identified through UNICEF’s IR projects, with a particular focus on routine and ZD immunization for children under five. The framework provided a structured way to map strategies against the core strategic level, such as political commitment, governance, sustainable financing, and community engagement, and operational levers, including workforce, infrastructure, supply chains, digital technologies, and monitoring systems [32]. This mapping method established a clear connection between implementation activities and measurable outcomes in both immunization and broader PHC system strengthening. It also allowed the extracted strategies to be linked directly to service delivery outcomes and policy-level changes, as documented in the literature [32,33,34,35,36,37,38]. Service delivery outcomes identified from the literature included increased vaccination coverage, stronger health systems, improved microplanning and data use, enhanced community trust, and better integration of immunization with other PHC services [33,35,36]. At the policy level, the strategies also fostered more integrated and resilient health systems, supported institutional learning, promoted health equity, reduced immunization gaps, empowered communities, and informed policy adaptations, especially in crisis-prone settings [33,34,35]. A summary of the framework is shown in Figure 1.

### 2.3. Identification of the IR Projects

The identification of projects followed a structured process. We searched the UNICEF IR website and online repository of IR projects [21], peer-reviewed publications from UNICEF and Gavi-supported Special Issues (*Journal of Global Health*—Pakistan Embedded Implementation Research for Immunisation Initiative (https://jogh.org/category/jogh/2021/jogh-collections/pakistan-embedded-implementation-research-for-immunisation-initiative/, accessed on 20 July 2025); *Health Research Policy and Systems*—Decision-Maker Led Implementation Research Supplement (https://health-policy-systems.biomedcentral.com/articles/supplements/volume-19-supplement-2, accessed on 20 July 2025); *Ethiopian Journal of Health Development*—Implementation Science Research in Ethiopia (https://ejhd.org/index.php/ejhd/issue/view/147, accessed on 20 July 2025; and others [39,40]), and directly consulted the UNICEF health team coordinating the IR projects and the team leads or principal investigators of the identified projects. Projects were included if they were completed, UNICEF-supported, and documented in reports, evaluations, or peer-reviewed publications with sufficient methodological detail. Ongoing or incomplete projects, materials without implementation detail, and opinion pieces lacking analytical depth were excluded.

### 2.4. Data Collection

Data collection was carried out through two complementary processes. First, systematic document reviews were undertaken to provide a comprehensive understanding of project implementation processes and outcomes. These reviews included program reports, evaluations, case studies, and monitoring data. Second, consultative validation discussions were conducted with team leads from selected IR projects. These discussions provided real-time interpretation, ensured that findings reflected on-the-ground realities, and helped to align the results with UNICEF’s strategic priorities. Projects were identified through a comprehensive search of the UNICEF IR website, published papers from UNICEF and Gavi-commissioned Special Issues, direct communication with members of the UNICEF PHC-HSS Unit supporting IR globally, and principal investigators of key projects. The inclusion criteria focused on documents providing substantive insights into implementation strategies, barriers, or facilitators, while materials such as short opinion pieces, documentaries, or those lacking analytical depth were excluded. Each IR project was examined in relation to relevant national and regional policies, strategies, and guidelines, paying particular attention to identifying implementation approaches and structural or operational factors contributing to success. Facilitators common across multiple contexts were highlighted, and findings were further contextualized through global and regional comparisons to identify lessons and opportunities for adaptation.

### 2.5. Data Extraction and Coding

All information and strategies were extracted verbatim and systematically recorded in an Excel database. Between 3 and 15 strategies were identified for each project, resulting in 326 strategies across all the reviewed documents. To strengthen methodological rigor, we developed a codebook aligned with the PHC framework, piloted it on three projects, and refined it through team discussion. Two reviewers were trained in its application, and 20% of the strategies were double-coded to assess consistency. Inter-rater reliability was measured using Cohen’s kappa [41], with discrepancies resolved through discussion until consensus was reached. The remaining strategies were coded independently, supported by regular team meetings to ensure consistency. Each strategy was mapped to one or more PHC levers, being classified by intervention type, operational focus, and outcomes. Where possible, strategies were linked to outputs, outcomes, and impacts, though the latter were more difficult to establish because most projects had not yet reached the five-year period typically required for long-term assessment.

### 2.6. Data Analysis

The data were analyzed using descriptive statistics, including frequencies and percentages, to summarize the distribution of strategies across projects and PHC levers. Inferential analyses such as *t*-tests or ANOVA were not applied, as the projects varied widely in design, scale, and context, limiting their comparability [42]. Instead, analysis emphasized pattern recognition, thematic synthesis, and cross-case comparison, which are widely used in evidence reviews of complex implementation research.

### 2.7. Consultative Validation

For the consultative validation, we considered a set of UNICEF-implemented IR projects across LMICs that were classified as successful based on pre-defined criteria such as reductions in the number of ZD children and scalability potential (Table 1). Team leads and at least one other implementer were invited to participate, of whom five responded (n = 5). Participants were contacted via email and provided with the questionnaire and research information sheet in advance, giving informed consent prior to interviews. Semi-structured interviews, guided by this study’s findings from the document reviews aligned with the conceptual framework, were conducted between 1 and 15 August 2025 via Zoom, Google Meet, or Teams. Each interview was conducted in English, lasted 45–60 min, and was audio-recorded using encrypted devices.

### 2.8. Ethical Consideration

This study relied on secondary analysis of existing project documentation and published materials. All IR projects had previously secured ethical approval or Institutional Review Board (IRB) clearance in their respective countries. These approvals are documented in journal Special Issues, including the *Journal of Global Health* (Pakistan Embedded IR), *Health Research Policy and Systems* (Decision-Maker-Led IR Supplement), and the *Ethiopian Journal of Health Development* (Implementation Science Research in Ethiopia). As this paper only analyzed previously approved and published data, no further ethical review was required. To complement the results of the secondary analysis, we conducted five consultations with IR team members. Although formal IRB approval was not obtained prior to the consultation, informed consent was secured from each respondent before the interview. This study adhered to established ethical standards for document-based implementation research, ensuring transparency, accurate representation of findings, proper citation, acknowledgment of originating institutions, respect for data ownership, and avoidance of misrepresentation [43].

## 3. Results

### 3.1. Characteristics of the Studies and the Intervention

We identified 36 IR studies (Appendix A), with 32 focused on the EPI [21,22,44,45,46,47,48,49,50,51,52,53,54,55,56,57,58,59,60,61,62,63,64,65,66,67,68,69,70,71], 3 focused on ZD children [72,73,74], and 1 addressing both domains in a cross-cutting manner [75]. These studies spanned 13 diverse LMICs, including Afghanistan [71], Bangladesh [75], the Central African Republic (CAR) [71], Chad [22,45,71], Ethiopia [22,48,50,59,60,61,62,63,64,65,66,67,68,69,70,71], India [22,47,71], Kenya [71], Kyrgyzstan [71], Madagascar [71], Myanmar [71], Nigeria [22,46,51,71], Pakistan [21,22,49,52,53,54,55,56,57,58,71,72,73,74], and Uganda [22,44,71] (Figure 2). Among the studies, 34 were country-specific [21,22,44,45,46,47,48,49,50,51,52,53,54,55,56,57,58,59,60,61,62,63,64,65,66,67,68,69,70,72,73,74,75], while 2 included multiple LMICs [22,71].

This broad geographic spread captured diverse health system contexts and immunization challenges, providing nuanced insights into delivery barriers and contextual adaptations. Interventions were categorized into 10 thematic intervention foci, covering 34 discrete intervention types/focus areas. As shown in Appendix A, the primary concentration of IR efforts was in immunization planning and delivery and data systems and decision support, with each factor comprising six interventions. These two domains accounted for the majority of activities, underscoring a strong programmatic focus on optimizing service delivery mechanisms and enhancing data-driven decision-making. Secondary areas of focus included demand generation and community engagement, vaccine hesitancy and acceptance, and supply chain and logistics. In contrast, fewer projects addressed governance and workforce, innovation and technology, research and evidence generation, emergency and conflict settings, and integrated approaches, revealing important gaps and highlighting opportunities for future targeted research and strategic investment.

### 3.2. Immunization Strategies Addressed in the IR Projects and the Outcomes

In this section, immunization strategies from diverse IR projects are mapped against the WHO’s 14 PHC levers to demonstrate how they align with core health system functions, offering insight into their role in enhancing immunization delivery and reaching ZD populations. A detailed analysis is provided in Appendix A.

The results in Figure 3 show a dominant focus on operational levers of PHC, which collectively account for 248 out of the 326 of the immunization strategies implemented in the countries studied (76.1%). Among these, the PHC workforce (n = 61, 18.7%) and monitoring and evaluation (n = 63, 19.3%) are the most frequently implemented, followed by models of care (n = 39, 12%) and digital technologies for health (n = 38, 11.7%). In contrast, the core strategic levers represent 78 of the 326 coded strategies (23.9%), with the engagement of communities and other stakeholders (n = 29, 8.9%) and governance and policy frameworks (n = 25, 7.7%) being the most prominent. Meanwhile, political commitment and leadership (n = 11, 3.4%) and funding and allocation of resources (n = 13, 4%) appear less frequently, suggesting that these foundational enablers may be less explicitly targeted or documented in programmatic strategies. Notably, engagement with private-sector providers and purchasing and payment systems were the least implemented operational levers, with only three (0.9%) and two (0.6%) mentions, respectively.

The approaches contributed to both service delivery and policy-level outcomes (Figure 4), with the most prominent outcome being the improvement of microplanning and data systems (n = 84, 23.5%), indicating the widespread use of digital innovations such as E-Vaccs, real-time dashboards, and GIS-enabled planning tools that enable precise targeting and agile decision-making. On the other hand, the enhancement of community engagement (n = 58, 16.2%) highlights the pivotal role of local actors, including religious leaders, health extension workers, and community committees, in building trust and increasing demand for vaccines through culturally tailored outreach such as SMS reminders and peer-support mechanisms. Fifty-six (15.7%) of the implemented strategies focused on strengthening health systems, reflecting concurrent investments in cold chain infrastructure, health workforce capacity-building, supervisory systems, and decentralized power sources such as solar energy for remote facilities. In total, 10.6% of the strategies (n = 38) increased vaccination coverage and were mainly focused on delivery innovations such as mobile vaccination units, extended service hours, and targeted outreach to underserved populations. As a policy outcome perspective, institutional learning (n = 27, 7.6%) was achieved through integrating IR findings and feedback loops to iteratively improve strategies. Moreover, only 19 (5.3%) of the strategies yielded policy integration and integration with primary health care services (n = 16, 4.5%), which revealed a growing alignment between immunization and broader health system strategies for enhancing efficiency and coherence.

Efforts to address health equity and reduce coverage gaps were also observed (n = 16, 4.5%), particularly through focused strategies for urban slums, nomadic communities, and conflict-affected areas. In contrast, sustainable financing models were the least commonly achieved goals (n = 12, 3.4%), indicating a gap in long-term funding strategies despite strong attention being paid to operational and technical enhancements. A detailed analysis is provided in Appendix A.

#### 3.2.1. Immunization Strategies Under Core Strategic Levers

##### Political Commitment and Leadership

*Strong governance, high-level political commitment, and community leadership have been central to advancing immunization strategies across LMICs*. In Chad and Nigeria, national and local authorities (government officials, traditional rulers, and community elders) actively promoted vaccine uptake by enforcing mandates, levying fines, and launching public campaigns aimed at building trust and compliance with immunization schedules [22,45,46]. These enforcement-based approaches were designed to strengthen both accountability and community participation.

*In Pakistan, a multi-tiered leadership structure proved critical.* The Ministry of National Health Services, Regulation, and Coordination (MoNHSRC) led national strategy development, supported by the Expanded Programme on Immunization (EPI) and the Polio Eradication Initiative (PEI), which jointly aligned with programmatic efforts [52,72]. Recognizing the interdependence between routine immunization and polio eradication, the Prime Minister’s Task Force elevated RI to a national agenda [72]. Respondents confirmed this: ‘*At that time*, *the federal EPI and the provincial EPI of Balochistan* … *it was very easy for us* … *the Ministry and Department of Health* … *they were on board* … *we had all of these stars in line and aligned*—*IR team Pakistan.’*

Leadership also extended into financing and operational management. In one province in Pakistan, despite resource constraints affecting the expansion of the Vaccine Analytics Network (VAN) [57], accountability was reinforced through direct financial commitment: ‘*We also put, after taking legal permission*, *the EPI manager of the province on payroll* … *he was a government officer* … *the program manager for the provincial EPI program*.—*IR team Pakistan*.’

*Participatory governance approaches were especially valued, fostering trust and contextual relevance*. As one respondent highlighted, ‘*It was a participatory approach* … *identify all stakeholders*—*officials in the health system or livestock system*, *but also the community* … *You let them lead it* … *Better that you leave this to those who are already there*.—*IR team Chad.’*

*At the community level, leadership was extended to tribal chiefs and religious figures*, *who played critical roles in driving mobilization in underserved areas* [55,58,71]. In Nigeria and Afghanistan, negotiations with non-state actors including armed groups enabled safe vaccine delivery [71]. In some regions, movement restrictions were used to enforce compliance [54].

Coordination across government sectors further strengthened leadership. As one key informant recalled, ‘*I had participants including the Secretary of Health, the Secretary of Finance … getting them to talk to each other about a joint mission* … *there is policy entrepreneurship and diffusion of innovation.*—*IR team Pakistan*.’ In Ethiopia, governance approaches emphasized data use, with decision-makers prioritizing data-driven strategies for enhancing immunization quality and responsiveness [50].

##### Governance and Policy Frameworks

*Alignment with national policy, community engagement, and structured coordination frameworks were central to advancing immunization strategies in LMICs.* In Uganda, public and NGO facilities demonstrated higher adherence to structured planning, with 71% and 60%, respectively, maintaining updated microplans, suggesting the effectiveness of organized service delivery mechanisms [44]. Across LMICs, the EPI served as a foundational tool, being used in 90.9% of projects to standardize immunization delivery [22]. Chad exemplified effective cross-sectoral collaboration, with the Ministry of Public Health partnering with the Ministry of Livestock through human–animal health services and a dedicated framework for nomadic health [22]. This echoed field-level experiences: ‘*You need adherence at the central level* … *but you also need involvement at the decentralized level*.—*IR team Chad*.’

*Decentralized and multi-level governance approaches further strengthened implementation*. In Nigeria, local health committees and subnational governments actively identified service gaps and drove community mobilization [51]. In Pakistan, decentralization was supported through a complex but effective governance model. A multi-tiered governance structure, leveraging high-level coordination bodies and inter-provincial committees [72], complemented by local enforcement strategies involving police, civil administrators [54], and engagement with religious leaders to strengthen community trust [73], enabled an agile response. Nationally, frameworks such as the National Emergency Action Plan (NEAP) and Gavi 5.0 guided the integration of PEI and EPI, underpinning a successful inactivated polio vaccine (IPV) rollout [73]. Emergency Operations Centers (EOCs) were established for coordination, and in insecure areas, vaccine delivery was secured through the support of security forces [72]. One respondent described the strength of embedding local actors: ‘*Having your own EPI people who are the beneficiaries* … *they are on board, they are asking for it and they are supporting it … that makes a huge difference.*—*IR team Pakistan*.‘ Over time, governance models evolved—Pakistan, for example, discontinued the use of coercive criminal intermediaries in favor of more transparent, community-driven approaches [57].

*However, governance gaps remained*. In Ethiopia, unclear supervisory mandates triggered revisions to governance protocols, though efforts to integrate SBCC, Standard Operating Procedures (SOPs), and job aids at the facility level showed promise [21,60,64,67]. At the same time, urban equity policies began to formalize: Pakistan introduced a dedicated urban immunization roadmap and countries such as Myanmar, Madagascar, and Nigeria integrated inclusive urban strategies into national frameworks [71].

##### Funding and Allocation of Resources

*A diverse mix of financing strategies underpinned immunization programs in LMICs*, *combining government allocations*, *donor support*, *and cross-sectoral contributions*. Chad, Ethiopia, India, Nigeria, Pakistan, and Uganda mobilized a blend of government allocations, donor contributions, and international agency support, particularly from GAVI and UNICEF, to underpin both service delivery and research activities [22,45,56]. Pakistan strategically restructured immunization budgets to support salaries, infrastructure, outreach, and community engagement, while procurement and logistics were centrally managed using pooled funds [49,55,72]. Salaries for EPI staff and transport were covered domestically, but human resource development remained largely donor-driven and underfunded [74].

*Sustainability was a recurring concern*. As one respondent warned, ‘*Still implementing it based on external funding … this is crucial for sustainability*. *You can’t sustain such activity based on 100% external funding—IR team Chad*.’ They further noted, ‘*Whenever any other country wants to do it*, *they need to put money on the table*, *in the budget*, *as part of district-level support*—*IR team Chad*.*’*

*Innovative cross-sectoral financing approaches were also evident*. For example, Chad’s implemented cost-sharing between public health and veterinary services, reinforcing One Health approaches [22]. By contrast, Ethiopia’s financial shortfalls left districts reliant on external resources to prevent stockouts [59]. Weaknesses included chronic vaccine shortages and limited funding for SBCC [60,68]. Equity issues arose where private providers charged fees while public services remained free, creating barriers for low-income households [68]. While some programs demonstrated financial flexibility, the respondents noted the need for economic evaluation in influencing policy: *‘*… *one can do a study of the economic analysis*—*how much cost is reduced and how many lives are saved*. *If something is published* … *you can affect the policy making*—*IR team Pakistan*.’

##### Engagement of Communities and Other Stakeholders

*Community engagement was a cornerstone strategy used across LMICs*. In Chad, Ethiopia, India, Nigeria, Pakistan, and Uganda, religious leaders, village chiefs, and local influencers promoted immunization in marginalized and hard-to-reach groups [22,71]. Chad relied on mobilization teams to reach nomadic populations, while Nigeria engaged CSOs, schools and culturally embedded communication channels [22,46]. Oversight committees, including ward development committees (WDCs), joint action committees (JACs), and social mobilization committees (SMCs), supported monitoring and accountability [51]. Pakistan institutionalized community participation by engaging TBAs as intermediaries between families and vaccinators [53,54,55,56]. Mosques, banners, and local forums were used for communication, complemented by slum health committees and loudspeaker campaigns targeting urban informal settlements [57,71].

Respondents stressed that engagement must address community needs beyond vaccination: ‘*We are offering something*, *one thing*, *but they’re demanding another thing* … *until and unless our immediate healthcare needs are met*, *we will not support your efforts to avoid a distant threat*—*IR team Pakistan*.’ Religious dynamics also required careful handling: ‘*Nobody produced any words from Quran or Hadith that prohibits vaccination … If something is coming out of a bearded man … you cannot just call it religious refusal.—IR team Pakistan*.’

Social listening emerged as a critical enabler: ‘*Most barriers are due to lived realities*, *not lack of awareness*. *Strategies must be adaptive and rooted in social listening*. *The zero-dose children are often ‘hiding in plain sight—IR team Kenya*.’ In Ethiopia, institutionalized community platforms such as the Women’s Health Development Team (WDT) and community scorecards were used to track immunization and newborn registration [59,60,64]. However, defaulter tracing and SBCC integration remained inconsistent. Male partners and religious figures were identified as crucial influencers, while traditional healers were less involved [62].

Engagement needed local tailoring: ‘*You need to ask the community what else they are interested in from health perspectives … clean water, snake bites*, *malnutrition* …—*IR team Chad*.” Education and feedback loops further strengthened trust: ‘*The community wanted to know the results*. *They came to our meetings and discussed things openly*’. Ultimately, empowering families, especially parents, to see value in immunization was key: ‘*Empowering the community is a benefit if the mother and the father think that there is benefit in immunizing their child*—*IR team Pakistan′*.

#### 3.2.2. Immunization Strategies Under Operational Levers

##### Model of Care

*Integrated and context-specific delivery models enhanced access and uptake*. In Uganda, structured microplanning improved service coordination and delivery [44], while Chad bundled human and animal vaccination campaigns with malaria prevention and antenatal services for nomadic groups [22]. Nigeria used schools and churches as vaccination points [46], while Ethiopia and Nigeria relied increasingly on fixed health facilities and inter-facility referrals [48,51].

Pakistan built strong community–health interfaces, with TBAs partnering with vaccinators in outreach camps, complemented by CHWs conducting door-to-door mobilization [53,54,55]. These efforts were reinforced through synchronized polio and routine immunization campaigns, particularly in high-risk regions [73]. Respondents underscored the importance of private-sector integration: ‘*If a baby is born in [the] private sector* … *if the insurance programs share the details of their baby with the EPI National Immunization Registry*, *then the national immunization people can follow up … gives a pinpointed identification of a potential missed or a potential zero-dose baby*.—*IR team Pakistan*.’ Regulatory enforcement was also proposed: ‘*If a private hospital wants to operate and do business*, *I will make it mandatory to my regulatory powers to have the vaccination services*.—*IR team Pakistan*.’

*Integration with maternal, child health*, *and WASH programs expanded access* [72]. Ethiopia’s health extension program ensured delivery in remote areas, shaped by women’s autonomy in health care decisions [62,65]. Equity-driven approaches included Ethiopia’s “Ketena” outreach [70], Pakistan’s mobile teams in urban slums [71], and extended service hours in Uganda and Kenya [71]. Humanitarian contexts also achieved high results, with Ethiopia’s IDP programs reaching 85.8% TT coverage in pregnancy and 76.5% full child immunization [59,62].

*Client-centered innovations were also seen in pain management and care delivery environments*. Health providers offered multiple pain-reduction strategies, both pharmacological and non-pharmacological: ‘*The providers proposed both pharmacological and non-pharmacological strategies … These included breastfeeding during the injection, skin-to-skin contact, and sugar solutions*.—*IR team Kenya*.’ Mothers were given agency: ‘*Mothers were allowed to choose the approach they felt most comfortable with*.’ However, cultural barriers remained: ‘*Some mothers were uncomfortable breastfeeding in such environments*, *so they opted for other methods*, *even if breastfeeding was known to be more effective in reducing pain—IR team Kenya*.’

##### PHC Workforce

*PHC workforce strategies centered on training*, *mobilization*, *and diversification*. Uganda trained managers and midwives in microplanning, complemented by peer-led workshops [44]. Chad and India relied on trained vaccinators, CHWs, midwives, accredited social health activists (ASHAs) and public health nurses to operationalize routine immunization [22,47,71]. Ethiopia deployed health extension workers (HEWs) to track households, stocks, and defaulters, supported by the Women Development Team [48,64,65], though refresher training gaps forced adaptive practices [60,66,69]. Male HEWs and vaccinators contributed to defaulter tracing and home-based services, reaching nearly one-third of postpartum mothers [62].

In Pakistan, cadres, comprising lady health workers (LHWs), lady health supervisors (LHSs), traditional birth attendants (TBAs), and male vaccinators, were mobilized for vaccine education, outreach, and administration. LHWs were described as ‘… *the backbone of primary care*—*IR team Pakistan*.’ The efforts of all the cadres were supported through induction training and consistent supervision [21,53,56,57]: ‘… *We organized the proper training of people* … *field team was trained*.—*IR team Pakistan*.’ Over 250,000 polio workers supported counseling, tracking, and logistics, reinforcing routine immunization [73].

*Workforce shortages highlighted urban–rural inequities*: ‘*I have 2500 positions for EPI vaccinators and half* … *are filled* … *mostly in the urban centers*. *And nobody’s willing to go to the rural area*.—*IR team Pakistan*.’ In response, health officials reportedly took assertive steps: ‘The health secretary said, give me their names. If they’re not willing to work, we’ll hire more people.’

Collaboration with private providers supported rapid mobilization: ‘*We partnered with both private and public primary healthcare providers* … *Because we already had strong working relationships with these networks*, *convening them to discuss next steps was straightforward*.—*IR team Kenya*.‘ However, implementation challenges persisted: ‘*Some procedures* … *were time-consuming and occasionally skipped in high-volume settings* … *smaller facilities managed better*, *and some even provided private breastfeeding spaces*.—*IR team Kenya*.’

Nigeria filled gaps with midwives, CHWs, and medical students [51]. Peer-support structures boosted morale in Afghanistan, Myanmar, and Nigeria and female vaccinators improved access in Afghanistan [71]. Continuous in-service training, including cold chain, requisition, and digital tools, remained central.

##### Physical Infrastructure

*Logistics and infrastructure strategies in LMICs prioritized innovative transport solutions*, *cold chain expansion*, *and facility-level upgrades to improve vaccine accessibility*, *safety*, *and system efficiency*. In Nigeria, CSOs employed motorcycles and boats to navigate hard-to-reach areas [46]. Pakistan invested in motorbikes and regular fuel supplies for vaccinators, enabling consistent outreach to remote populations, while cold chain infrastructure was strengthened at both federal and provincial levels to safeguard vaccine integrity across the supply chain [57,72]. Because of conflict and rugged terrain, Ethiopia transported vaccines via military convoys from central facilities to health posts [59,65].

*Facility-level improvements further reinforced cold chain reliability and service readiness across countries*. In Ethiopia, much functional cold chain equipment (refrigerators, thermometers, cold boxes, and ice packs) was used [67]. Solar-powered systems were extensively used in remote or off-grid regions across Ethiopia, Chad, and Nigeria, providing a sustainable energy solution for vaccine storage and transport [67,71], and waste management was strengthened with incinerators and safe disposal system [61]. These gains improved access: in Ethiopia, 88.2% of mothers reached immunization sites within 30 min [62].

##### Medicines and Other Health Products to Improve Health

*Immunization logistics systems were strengthened through microplanning*, *inventory management*, *and safety protocols*. Uganda relied on facility-level forecasting, enabling timely requisition and reducing the likelihood of stock-outs [44], while Nigeria improved last-mile vaccine availability [51]. Ethiopia emphasized quality control and safe handling, where vaccines were properly labeled and administered using auto-disable (AD) syringes, and cold chain tracking to safeguard vaccine potency throughout the distribution network [61]. Key Respondents emphasized bundling services: ‘*we combined vaccination with vitamin A*, *antiparasitics*, *nutritional biscuits*, *treated bed nets* …’ and ‘*whenever they have other issues and then you take it seriously … they will accept your vaccination strategy very smoothly*.—*IR team Chad*.’

##### Engagement with Private-Sector Providers

*Efforts to engage private-sector providers in immunization remained limited, highlighting missed opportunities to expand service delivery and strengthen system integration*. In Ethiopia, private health facilities provide immunization services on a modest scale, constrained by structural, logistical, and operational barriers [68]. While these facilities often depend on vaccines supplied by the government, challenges in maintaining adequate cold chain storage frequently arise, potentially jeopardizing vaccine efficacy [68]. Private providers were minimally integrated into the national immunization data reporting systems, weakening monitoring and accountability efforts [68].

##### Purchasing and Payment Systems

*Incentive-based purchasing and payment systems were strategically employed to enhance the engagement of informal health actors in immunization delivery*. In Pakistan, TBAs were mobilized to support immunization efforts, particularly during polio campaigns, through a structured incentive model that combined stipends with non-monetary rewards [53]. These mechanisms not only acknowledged and sustained TBA motivation but also capitalized on their deeply rooted trust within households to extend vaccine outreach and acceptance in underserved communities [53].

##### Digital Technologies for Health

*Digital strategies advanced immunization planning and monitoring*. Pakistan’s electronic vaccine registration system (e-Vaccs) tracked vaccinator performance, absenteeism, and service coverage with GPS and real-time updates such as defaulter alerts, enhancing outreach efficiency [21]. Integrating the Vaccine Analytics Network (VAN) with the Vaccine Logistics Management Information System (vLMIS) improved supply chain transparency and reporting [57].

*Communication technologies complemented these systems*. As one respondent explained, ‘*we used to send them periodic SMSs and IVRs* … *the vaccination date is near*, *so they get an SMS and as well as an IVR*.—*IR team Pakistan*.’ These mobile innovations, including SMS reminders, IVR calls, and AI-enabled messaging, helped caregivers, particularly fathers, to stay engaged and counter misinformation [21,57].

*Digital logistics tracking systems ensured vaccine availability and continuity*. One respondent emphasized, ‘… *digitally enter and keep track* … *when one of the vaccines was getting shorted*, *something would pop up in the system*—*IR team Pakistan′*, illustrating how automated alerts helped to avert shortages and sustain continuity in immunization programs. The broader feedback echoed this perspective: ‘*Instead of a paper system*, *a digital system works well* … *tracking vaccine supply*, *alerting shortages, and maintaining continuity*—*IR team Pakistan*.’

Other digital innovations included QR code tracking for individual child immunization in Pakistan and the use of WhatsApp groups in Pakistan and Afghanistan for real-time coordination and updates [71]. In Ethiopia, digital dashboards and visualized facility-level data supported decision-making, while standardized immunization registers, e-vaccine supply chain pilots, and platforms such as mBrana facilitated stock management [65,67,70]. Uganda enhanced microplanning through data from outreach logs and demographic profiling [44], and India complemented digital interventions with traditional media campaigns to promote vaccine messaging [47]. Additionally, multiple countries, including Ethiopia, India, and Uganda, deployed integrated health information systems such as the electronic Community Health Information System (eCHIS) to track vaccine stocks and delivery services [71].

*The integration of data systems emerged as a promising area for improved coordination.* As one key respondent noted, ‘*Make your claims management processing digital … One integration that came forward was the integration of the insurance claims database and the national immunization registry* …’ This integration helped to trace children born in private health facilities and enabled follow-up: ‘*So that the EPI program knows very well* … *which child was born in a private center and where do they live so that they can cover those children during their outreach activities*—*IR team Pakistan*.’

##### Systems for Improving the Quality of Care

*Efforts to improve immunization service quality included planning*, *supervision*, *and data management strategies*. In Uganda, routine reviews of microplans allowed for the timely adaptation of outreach strategies, ensuring responsive service delivery aligned that was with evolving community needs [44]. These planning efforts were further reinforced via targeted awareness campaigns in India that aimed to dispel misinformation and promote vaccine acceptance [47]. In Pakistan, collaboration between immunization teams, lady health workers (LHWs), and polio programs enhanced coordination and information flow, strengthening overall immunization performance [49].

Ethiopia adopted integrated supervision checklists embedded with data quality metrics, enabling structured performance evaluations at the facility level [50]. Similarly, in Pakistan, regular joint meetings across teams supported collaborative problem-solving and improved inter-program alignment [53]. However, challenges remained in both countries, with supervisory coverage reported as inconsistent in some areas, reflecting the need for the better implementation of oversight plans and accountability mechanisms [74].

*Immunization safety and cold chain integrity were advanced through robust clinical and logistical protocols*. In Ethiopia, adherence to safe injection practices was widespread, with health workers routinely checking fridge tags twice daily to maintain appropriate vaccine storage conditions [61]. Facilities also regularly employed vaccine vial monitoring (VVM) to verify vaccine quality and safety [67]. Additionally, contingency plans for managing cold chain emergencies were displayed in several health facilities, demonstrating preparedness for unexpected disruptions [67]. The key respondents validated these developments: ‘*Measles take-up also increased* … *the delay was shortened* … *Before*, *people used to miss their children’s upcoming vaccination*. *Now they were reaching out for the vaccination—IR team Pakistan*.’

##### Primary Health Care-Oriented Research

*Participatory action research emerged as a core strategy for strengthening immunization programs through inclusive*, *evidence-based approaches*. In Chad, Ethiopia, India, Nigeria, Pakistan, and Uganda, research projects combined participatory methodologies with tailored communication strategies to expand service reach and improve relevance. These efforts incorporated qualitative tools, including key informant interviews, focus group discussions, and in-depth interviews, to capture local insights, contextualize challenges, and adapt interventions accordingly [22]. A key component of this strategy was the direct involvement of decision-makers in research processes, which ensured alignment with national priorities and facilitated the integration of findings into health policy and programmatic decisions [22].

The institutionalization of participatory research approaches was notably evident in Pakistan, where EPI implementers were designated as principal investigators to enhance the ownership and operational applicability of research activities [52]. This collaborative model encouraged joint priority setting between EPI managers and researchers, resulting in studies that addressed key implementation bottlenecks. To promote uptake of findings, results were disseminated through national workshops involving policymakers and implementers [52]. Research also focused on practical concerns, such as the quality of supervision and the feasibility of service delivery models, by engaging district-level EPI stakeholders in the co-design of strategies to inform scale-up [21]. In Ethiopia, parallel use of qualitative methods allowed for documenting specific barriers to vaccination among hard-to-reach populations, providing actionable evidence to guide future program investments [63].

##### Monitoring and Evaluation

*Monitoring and evaluation strategies were central in strengthening immunization systems by enhancing data accuracy*, *service accountability*, *and targeted delivery.* In Uganda, the foundation for effective monitoring was evident, with 95% of facilities utilizing health information systems and routinely reviewing microplans to track immunization coverage and identify defaulters [44]. Chad and Ethiopia employed mixed-method evaluations and regular review meetings to assess campaign outcomes and detect service gaps [22,45]. Ethiopia’s approach included Performance Monitoring Teams (PMTs), data verification protocols, and transparent sharing of performance metrics, empowering frontline workers to make informed, evidence-based decisions [50]. Similarly, in Nigeria, the National Health Management Information System (NHMIS) enabled improved tracking of vaccine-seeking behavior and programmatic planning [51].

*The digital integration of monitoring tools also advanced immunization oversight*. In Pakistan, systems such as E-Vaccs and district-level dashboards facilitated real-time monitoring of vaccinator activity and defaulter tracking, hence improving data accountability and program responsiveness [21,57]. VAN-supported stock management and the consolidation of polio and EPI datasets allowed for identifying ZD children, though persistent system linkages posed challenges [73]. Supervisory improvements, including redesigned checklists and scheduled oversight visits, contributed to data reliability and service quality [21]. In Ethiopia, strategies such as the use of vaccine vial monitors (VVMs), manual ledger entries, and verification through BCG scars and immunization cards, along with community validation, ensured quality control and uncovered inconsistencies [61,69]. One key respondent described how qualitative methods added value: ‘*We used a self-reporting tool for mothers … Our research team analyzed these reports across different cohorts to determine the effectiveness of the interventions*.—*IR team Kenya*.’

*Geospatial innovations also supported equity-focused planning*. Real-time GIS-based microplanning was deployed in Afghanistan, Pakistan, Kenya, and Myanmar to optimize vaccine outreach in underserved regions [71]. Meanwhile, Ethiopia’s rollout of DHIS2 enabled digitized reporting and increased alignment with national systems. However, not all data systems were inclusive. As one key informant respondent, ‘*Most of the data systems concern sedentary populations … nomadic communities are not involved*’ and to address this, efforts were made to align data with mobility patterns: ‘*You link the information about the nomads’ routes and timing with health centers … so you don’t lose much follow-up*—*IR team Kenya*.’

## 4. Discussion

This study examined UNICEF-supported IR strategies for immunization through the PHC lever framework, which provided a structured approach to unpacking system-level enablers of progress toward reaching ZD children. Using the PHC levers framework, the analysis systematically mapped IR strategies to core and operational domains, providing a structured way to assess their contributions. In doing so, this study offers a novel lens through which to understand the complexity of interventions across the critical dimensions of the PHC framework. By situating interventions within both core and operational domains, the analysis highlights that equity challenges in immunization are not the result of isolated technical gaps but reflect systemic weaknesses across the broader PHC ecosystem. This framing resonates with global evidence emphasizing the need for integrated rather than fragmented approaches to immunization system strengthening [9,15]. The mix of service delivery innovations, workforce mobilization, digital tools, and participatory approaches observed here reflects an evidence-based pathway for building resilience and equity, aligning with the priorities set in the WHO’s Immunization Agenda 2030.

The findings highlight a strong programmatic focus on the organization, delivery, and monitoring of immunization services, showcasing how operational levers, such as workforce development, care models, and performance monitoring, drive on-the-ground implementation. This emphasis on the *‘how’* of delivery reflects a robust operational foundation, yet it also exposes underleveraged areas such as political commitment and research, which are crucial for long-term sustainability and system transformation. Strategies combining services delivery with governance and community engagement had the greatest impact, as demonstrated by the integration examples discussed earlier.

Importantly, these implementation strategies contributed meaningfully to outcomes at both the service delivery and policy levels. Digital tools enhanced microplanning and accountability, facilitating real-time targeting and coordination across multiple settings. Community engagement strategies, involving local leaders and health volunteers, directly increased vaccine demand and uptake, reinforcing the importance of culturally responsive service models. On the policy front, integrating IR and feedback loops fostered institutional learning, enabling iterative improvements in program design. However, despite these gains, areas such as sustainable financing and policy integration remain limited, signaling a critical need for strategic investments to ensure the coherence and durability of immunization efforts across health systems. The PHC lever framework, thus, emerges not only as a diagnostic and planning tool but also as a practical blueprint for scalable, equitable immunization delivery in fragile and underserved contexts.

However, persistent challenges remained. Heavy reliance on political mandates and enforcement measures, such as movement restrictions and fines, produced short-term gains but risked eroding trust, autonomy, and voluntary health-seeking behaviors, raising ethical concerns about consent and public confidence [22,46]. Moreover, dependence on short-term political cycles and donor agendas, rather than sustained domestic financing or institutional reforms, underscored the fragility of these approaches and highlighted the need to balance top-down political support with bottom-up community ownership.

Our findings highlight that coordination across governance levels was central to sustaining immunization outcomes, echoing evidence from other LMIC contexts where multi-tiered systems linking national commitment with local delivery enhanced efficiency and accountability [76,77,78]. Similarly to studies on health system resilience, mechanisms such as EOC and One Health collaborations functioned as adaptive platforms that integrated resources, reduced duplication, and enabled rapid responses to emerging challenges. At the community level, alignment with sociocultural norms and the engagement of traditional authorities reinforced legitimacy and trust, consistent with the literature showing that locally grounded governance models outperform enforcement-driven approaches that erode voluntary health-seeking behavior [76,77].

At the same time, our results underscore challenges widely documented in the literature regarding weak political consistency and the fact that fragile financing undermines sustainability. As highlighted in comparative analyses of health financing reforms, dependence on short-term donor cycles constrains the scaling-up and institutionalization of innovations [78]. Similarly, reliance on temporary negotiations with non-state actors has been shown to offer only short-lived gains unless paired with investments in trust-building and ethical safeguards [77]. These findings suggest that coordination in immunization cannot be viewed narrowly as administrative alignment; rather, it requires participatory, rights-based governance and stable financing to ensure that lessons from successful pilots translate into durable and equitable system strengthening.

Our analysis shows that delivery innovations, such as cold chain technologies, mobile clinics, drones, and service integration, fit within a broader body of evidence demonstrating the importance of context-responsive approaches in fragile and underserved settings. Similarly to findings from Bangladesh and India, maternal autonomy and multisectoral entry points (linking immunization with maternal health, nutrition, and WASH) were central in driving uptake [79]. Evidence from humanitarian emergencies also supports our observation that flexible delivery mechanisms, including strengthened transport and power solutions, sustain high coverage among displaced populations [80]. These results align with global discussions on the value of using community-led approaches over coercive strategies, reinforcing research conducted by others showing that participatory engagement and localized governance yield more durable outcomes [76,81]. At the same time, the increasing use of real-time data and digital tools for microplanning reflects convergence with emerging global trends in precision public health, though the literature cautions that such tools require parallel investment in workforce capacity and feedback systems to achieve full impact [82].

At the structural level, our findings echo earlier studies highlighting persistent supply chain vulnerabilities and the limited role of private-sector providers in low- and middle-income countries, where public–private partnerships remain uneven [83,84,85,86,87]. Incentive-based models, including non-monetary rewards for community health actors, also resonate with broader evidence demonstrating how such strategies can strengthen local delivery and community trust [88,89,90].

Equity remained a cornerstone across delivery models, aligning with wider evidence that pro-poor policies and strong regulatory frameworks are essential to prevent exclusion in mixed public–private systems [91]. Our findings suggest that adaptive, context-driven approaches that integrate services and build on community trust can extend coverage to underserved groups, consistent with experiences in Afghanistan and Nepal where community-embedded models sustained service continuity [92,93]. The expansion of immunization through alternative delivery sites, such as schools, churches, and community centers, reflects evidence from community-rooted approaches that emphasize trust as a central driver of uptake [94,95]. Importantly, demand generation and social listening extended beyond message content. Echoing the findings of Del Riccio and Smith, vaccine acceptance was strongly shaped by alignment with trusted information channels, whether institutional, social, or interpersonal, rather than message content alone [96,97]. This perspective helps to explain why Pakistan’s federal social media campaigns and Ethiopia’s community-driven platforms proved effective, reinforcing the need to prioritize channel legitimacy alongside content delivery.

Digital innovations have become increasingly central to immunization system strengthening, consistent with the literature highlighting their role in real-time transparency, supply chain management, and accountability [98]. Tools such as registries, SMS reminders, mobile applications, and dashboards allowed frontline workers to monitor children and vaccine stocks more effectively, thereby reducing the number of missed opportunities. Evidence from Pakistan and Nigeria illustrates how dashboards and geo-mapping supported targeted outreach to hard-to-reach and urban informal settlements, while similar results from other African contexts demonstrated comparable gains in equity and efficiency [98]. At the same time, critiques in the literature note that digital tools require parallel investments in workforce capacity, digital literacy, and adaptive feedback mechanisms to achieve lasting impact [79]. Quality improvement measures, combined with participatory action research, further strengthened program adaptability, echoing evidence that embedding implementers as research leaders promotes ownership and evidence-informed policy translation. Finally, the advances in monitoring and evaluation documented here align with global shifts toward precision public health, enabling more responsive, data-driven immunization systems. These insights show that, while digital and organizational innovations are powerful enablers, their effectiveness is contingent on their integration with broader system reforms, equity strategies, and sustainable governance structures [91,92,93,94,95,96,97,98,99,100,101].

### 4.1. Limitations

This study has some limitations. First, the analysis relied primarily on UNICEF-supported IR projects. While these projects provided valuable insights into strategies for overcoming immunization barriers, they may disproportionately represent positive or successful experiences, as unsuccessful or discontinued projects were not systematically included. This reliance may have introduced the risk of selection bias and may limit the extent to which failures, challenges, or unintended consequences are fully captured. Second, the findings may not be entirely generalizable to non-UNICEF contexts or other LMICs, given that UNICEF-supported projects often benefit from technical assistance, financial support, and established networks that may not be available elsewhere. Although many of the lessons are transferable, the degree to which they can be applied in settings without similar institutional backing requires further exploration.

Third, the study’s conclusions about the mechanisms of effectiveness and contextual factors influencing outcomes are based on descriptive statistics, project documentation, and self-reported data. While this approach enabled a broad synthesis across diverse contexts, it also makes the findings subject to reporting bias and limits our ability to draw causal inferences. The absence of inferential statistical analyses (such as *t*-tests or ANOVAs) and control groups was not possible, as it was a review of evidence, which may further constrain our capacity to isolate the impacts of specific strategies from external influences, including policy changes, epidemics, or broader system reforms. Fourth, although the study covered 13 countries, the distribution of projects was uneven, with countries such as Pakistan and Ethiopia contributing disproportionately large numbers of cases. This imbalance may reduce external representativeness and limit the generalizability of the findings to other LMICs with different sociopolitical, economic, or health system conditions.

Finally, while this study highlights innovative strategies such as digital health tools, participatory approaches, and multisectoral coordination, it does not fully disentangle how these mechanisms interact with contextual factors, such as governance quality, financing models, or community trust, to shape outcomes. Future research should, therefore, employ more rigorous mixed-method designs, including comparative case studies with counterfactuals, to better assess causality and strengthen the evidence base for scaling these strategies across diverse settings

### 4.2. Policy Implication for Eliminating ZD Children

The findings underscore critical policy implications for advancing strategies that can reach ZD children and strengthen equitable immunization. While interventions improved overall coverage, they also created entry points for reaching children who had never received vaccines, even when not explicitly designed for ZD populations. This highlights the importance of embedding ZD targeting explicitly into national immunization strategies, ensuring that programs move beyond aggregate coverage targets toward equity-focused approaches that disaggregate data, align service delivery with community needs, and tailor interventions for marginalized groups.

Realizing these goals requires clarity on institutional roles, sustainable financing, and the systematic use of IR. Ministries of health should take the lead in institutionalizing IR within national immunization and PHC programs, ensuring that evidence informs annual planning and routine practice. Donors and partners such as Gavi, WHO, and UNICEF can provide catalytic financing and technical support to scale successful models, while NGOs and civil society organizations play an essential role in strengthening demand generation through culturally trusted, community-driven approaches.

Financing remains a critical barrier. National governments must prioritize domestic resource mobilization and integrate ZD-focused strategies into broader health financing reforms and UHC schemes. Innovative approaches, such as blended financing, results-based funding, and sectoral cost-sharing, can complement donor support and sustain efforts in hard-to-reach populations. Dedicated streams for last-mile delivery, community engagement, and digital health systems are especially vital for ensuring continuity.

Finally, scaling and institutionalizing IR requires formal feedback loops between research, implementation, and policy. Embedding findings into sector-wide reviews, monitoring systems, and coordination platforms can improve learning and adaptation. Regional and global knowledge-sharing mechanisms should also be strengthened to accelerate cross-country learning and help with tailoring approaches to fragile and resource-limited settings. By coupling national leadership and financing commitments with locally participatory approaches, governments and partners can ensure that ZD children are systematically identified, tracked, and reached, thus advancing the goals of the Immunization Agenda 2030.

## 5. Conclusions

This study highlights the critical importance of context-adapted, system-oriented strategies for advancing equitable immunization coverage, particularly for ZD children in underserved settings. By applying the PHC levers framework to IR, we provide a structured way to analyze how governance, service delivery, community engagement, financing, workforce, and data systems interact to influence immunization outcomes. Importantly, IR offers distinct advantages over traditional health system research. Whereas conventional approaches often focus on static assessments or retrospective evaluations, IR emphasizes the delivery processes, generating real-time, actionable evidence that guides program adaptation and scale-up. This practical orientation enables IR to bridge the gap between policy and implementation, ensuring that interventions are responsive to local contexts and more likely to achieve sustainable impact. Closing persistent coverage gaps will, therefore, require investments that extend beyond vaccines—strengthening PHC systems, securing sustainable financing, and institutionalizing IR as a routine component of health programming. Embedding IR within national strategies creates pathways for continuous learning and course correction, positioning it as an indispensable driver of equity-focused transformation. As countries pursue the Immunization Agenda 2030 and the Sustainable Development Goals, harnessing IR-informed approaches will be essential for ensuring that no child is left behind, regardless of geography, gender, or circumstance.

## Figures and Tables

**Figure 1 vaccines-13-01040-f001:**
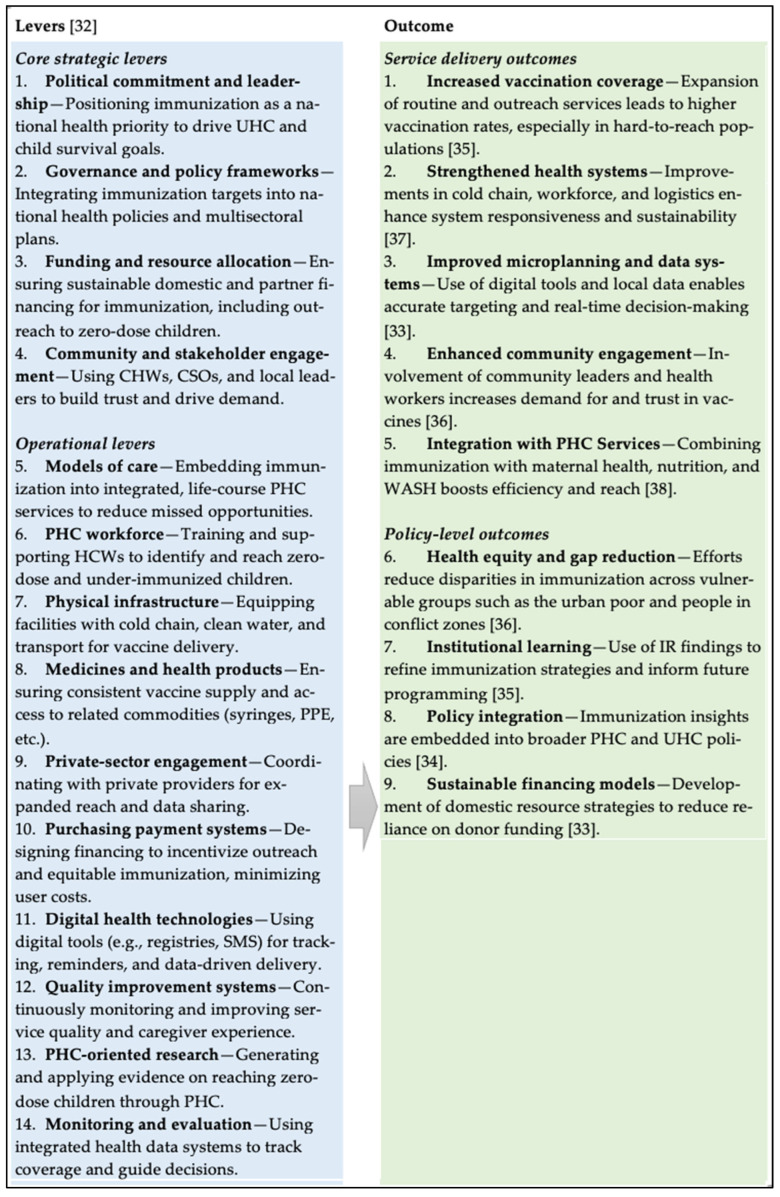
Analytical framework (source: Authors’ adaptation from WHO-UNICEF PHC framework and other literature reviews [32,33,34,35,36,37,38]).

**Figure 2 vaccines-13-01040-f002:**
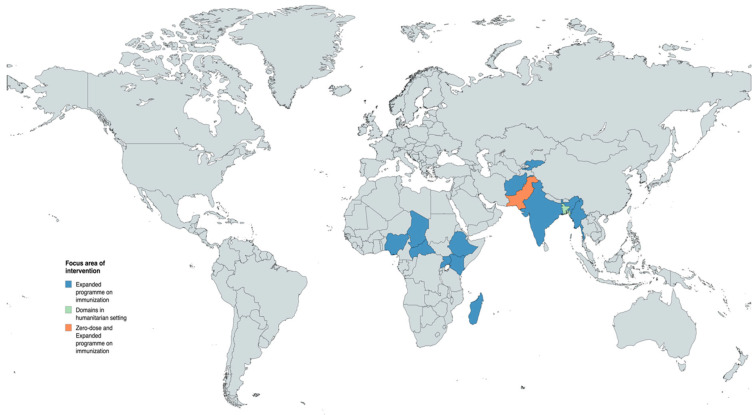
Countries and areas of intervention.

**Figure 3 vaccines-13-01040-f003:**
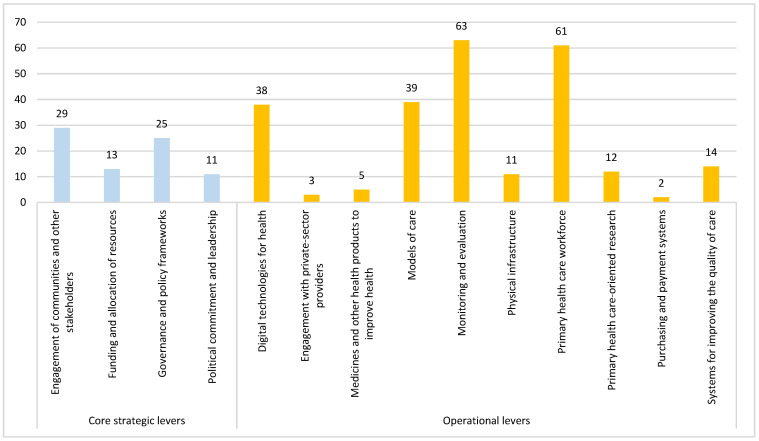
The classification of the immunization strategies based on the PHC framework.

**Figure 4 vaccines-13-01040-f004:**
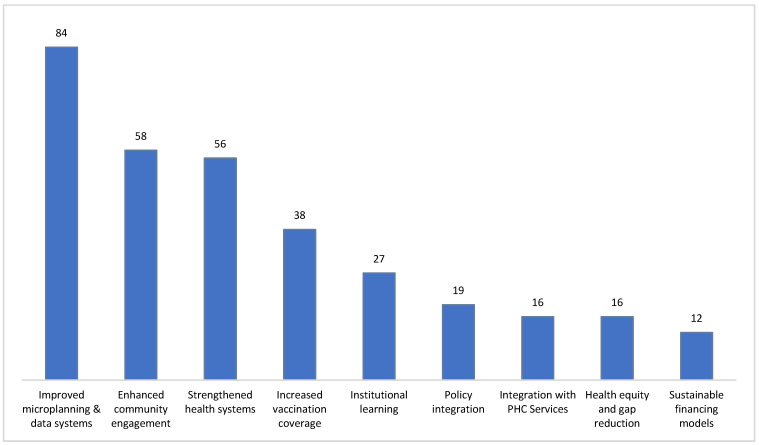
Service delivery and policy-level outcomes.

**Table 1 vaccines-13-01040-t001:** Studies considered for in-depth consultative discussion.

Changes Made	Focus Areas	Country
* Improved community mobilization and communication strategies in two wards to increase immunization coverage	EPI	Nigeria
m-Health technology and recruitment of new vaccinators introduced	EPI	Pakistan
Pain management strategy scaled-up in several hospitals	EPI	Kenya
* Birth registration enhanced through inclusion in MNCH and EPI programming and in community health platforms	BR4MNCH	Ethiopia
Federal government involved in social media content management to reduce anti-vaccine propaganda	EPI	Pakistan
* Improved vaccine literacy by creating a manual for health workers to build their knowledge and to better respond to the vaccine misinformation circulating in the communities	EPI	India
Enhanced communication strategies for reaching nomadic communities for vaccination through integrated human and animal vaccination campaign	EPI	Chad
Improved supply-side service delivery by provincial authorities	EPI	Pakistan

* Note: studies where respondents did not participate in the in-depth discussions although reached.

## Data Availability

All data are available within the article and the accompanying Appendix A.

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
