# Peer review of "Strengthening Primary Health Care Through Implementation Research: Strategies for Reaching Zero-Dose Children in Low- and Middle-Income Countries’ Immunization Programs"

_vaccines, 2025, doi:10.3390/vaccines13101040_

Round 1

Reviewer 1 Report

Comments and Suggestions for Authors

Summary
This manuscript compiles 36 UNICEF-supported implementation research (IR) projects across 13 LMICs, mapping 326 strategies to WHO/UNICEF primary health care (PHC) levers: it identifies recurrent approaches, such as monitoring & evaluation, workforce strengthening, digital tools, while highlighting micro-planning and community engagement gains, though governance and sustainable financing remain limited.

Some issues should be addressed before the manuscript can be considered for publication:

  • The portfolio is composed only of “completed and successful” IR projects, without systematic search or inclusion of unsuccessful experiences: this selection bias may undermine generalizability, therefore i suggest describing a reproducible identification process, clarify exclusion criteria, and reflect critically on the implications for bias.
  • The extraction of 326 “strategies” lack detail on coder training, development of a codebook, or inter-rater reliability. Mapping to PHC levers remains subjective. Considered this, authors should provide methodological safeguards (e.g., double coding, kappa scores) or acknowledge the interpretive limits.
  • Reported results are dominated by outputs (“improved microplanning,” “enhanced engagement”), with long-term impacts explicitly speculative. For credibility, authors should adopt standardized outcome metrics (coverage, equity gaps closed, cost-effectiveness) and temper causal claims accordingly.
  • Although five semi-structured, audio-recorded interviews were conducted, the manuscript states that no ethics approval was required. At a minimum, an institutional review board exemption or prior ethical clearance should be documented. Without this, the ethical foundations remain insufficient.
  • Numerous “Error! Reference source not found” placeholders and missing cross-references weaken readability. Moreover, the claim of being the “first” application of PHC levers to IR is unconvincing without engaging with prior PHC-equity or IR synthesis literature. Authors must situate their contribution more modestly.
  • When discussing demand generation and social listening, the authors should not only emphasize what messages were delivered but also how and through which channels communities access and trust information. Evidence shows that refusal and acceptance are patterned less by message content than by alignment with preferred information sources (institutional, social, or interpersonal). Integrating this concept would strengthen the analysis of Pakistan’s federal social media interventions and Ethiopia’s community platforms, showing why delivery channels matter as much as content (Del Riccio 2022, Smith 2017).

Refs

  • Del Riccio M, Bechini A, Buscemi P, Bonanni P, On Behalf Of The Working Group Dhs, Boccalini S. Reasons for the Intention to Refuse COVID-19 Vaccination and Their Association with Preferred Sources of Information in a Nationwide, Population-Based Sample in Italy, before COVID-19 Vaccines Roll Out. Vaccines (Basel). 2022;10(6):913. Published 2022 Jun 8. doi:10.3390/vaccines10060913
  • Smith LE, Amlôt R, Weinman J, Yiend J, Rubin GJ. A systematic review of factors affecting vaccine uptake in young children. Vaccine. 2017;35(45):6059-6069. doi:10.1016/j.vaccine.2017.09.046

Author Response

Authors have addressed all the comments raised by the reviewers and submitted a revised version of the manuscript with point by point response sheet for the reviewers

Reviewer 2 Report

Comments and Suggestions for Authors

In vaccines-3874424, Oyugi et al. present strategies to strengthen primary health care and reach zero-dose children in LMIC immunization programs. The topic of this manuscript is interesting and fits well the scope of Vaccines. The reviewer feels it can be accepted after some minor amendments.

(1) Reliance on UNICEF-supported projects may favor positive outcomes, while failures or challenges are under-discussed. The authors have to discuss this point.

(2) The applicability of these findings to non-UNICEF contexts or other LMICs is not sufficiently elaborated. The authors have to discuss this point.

(3) Please expand the discussion of mechanisms of effectiveness and contextual conditions that influenced outcomes.

(4) Please elaborate on the transferability of lessons learned to settings outside the included UNICEF projects.

Author Response

Authors have addressed all the comments raised by the reviewers and revised the manuscript accordingly. Please find enclosed draft word document with point by point response to the reviewers.  

Reviewer 3 Report

Comments and Suggestions for Authors

This article systematically analyzes 36 implementation research projects supported by UNICEF, for the first time applying the PHC Levers framework to identify key immunization strategies and their impact on zero-dose children coverage. The conclusions have certain policy implications. However, there is insufficient transparency in the methods section, and the statistical analysis methods are relatively simple, with some potential selection bias and reporting bias.

1.The methods section describes the data collection, coding, and analysis process, but lacks specific details such as tool versions, coding manuals, and interview guides, which makes it difficult to fully replicate.

2.Primarily descriptive statistics (frequencies, percentages) were used, without inferential statistics (such as t-tests/ANOVAs), which limited the ability to make causal inferences.

3.Some conclusions (such as 'strategy effectiveness') rely on self-reported data and project documentation, which may be subject to reporting bias.

4.No control group, making it impossible to rule out the influence of external factors such as policy changes or pandemics.

5.The sample lacks representativeness, although it covers 13 countries, some nations (such as Pakistan and Ethiopia) have too many samples, which may affect generalizability.

Author Response

(The authors gave the same response as above.)

Reviewer 4 Report

Comments and Suggestions for Authors

The manuscript addresses a highly relevant and timely issue-the persistent challenge of zero-dose children in low- and middle-income countries (LMICs). By framing this problem within the context of primary health care and implementation research, the paper contributes to both immunization and health systems strengthening discourses. The focus on strategies to identify, reach, and sustain vaccination in these children aligns with global goals such as Immunization Agenda 2030 and Universal Health Coverage.

The manuscript is generally well-structured, clear, and policy-oriented, making it suitable for a journal like Vaccines. However, several areas would benefit from deeper conceptual grounding, methodological clarity, and integration of empirical evidence.

Some comments for improvements:

  1. The abstract would benefit from including key quantitative insights or at least synthesized findings, rather than only general statements.
  2. Introduction. The introduction effectively sets the stage but could be improved by providing recent statistics on zero-dose children (e.g., UNICEF/WHO 2023 estimates: ~14.3 million zero-dose children globally).
  3. Methods. The methodology underlying the synthesis is unclear—was this a narrative review, scoping review, or policy analysis? Clearer methods would enhance scientific rigor.
  4. Discussion. Could expand on the role of digital health, innovations in vaccine delivery, and intersectoral collaboration (education, social protection, gender empowerment).
  5. Conclusion. Strong, but could better highlight the novelty of implementation research compared to traditional health systems research.
  6. While the manuscript highlights the importance of implementation research, the policy recommendations are somewhat generic. They could be strengthened by:

    -Identifying which actors (ministries of health, donors, NGOs) should implement specific strategies.

    -Suggesting how implementation research can be scaled or institutionalized within immunization programs.- Addressing financing mechanisms, which are a critical barrier in LMICs.

Author Response

(The authors gave the same response as above.)

Round 2

Reviewer 1 Report

Comments and Suggestions for Authors

I thank the authors for incorporating my comments in the manuscript. 

Reviewer 3 Report

Comments and Suggestions for Authors

After carefully reading the author's response letter, I found that although the authors did not make fundamental changes to the statistical methods and research design, they effectively addressed my core concerns by enhancing the transparency of the methods section and thoroughly and honestly discussing the limitations of the study. While there are still some minor imperfections, it is acceptable.

Reviewer 4 Report

Comments and Suggestions for Authors

I have carefully reviewed the revised version of the manuscript. The authors have thoroughly addressed all the issues raised in the previous review, including clarifications in the methodology, strengthening of the discussion, and refinement of the conclusions. The revisions have significantly improved the clarity, coherence, and overall quality of the paper.

The manuscript now provides a well-structured and comprehensive analysis of implementation research strategies to strengthen primary health care and to reach zero-dose children in LMIC immunization programs. The arguments are clearly presented, and the paper makes a valuable contribution to the field of global health and immunization research.